# Forward variable selection enables fast and accurate dynamic system identification with Karhunen-Loève decomposed Gaussian processes

Kyle Hayes[1,2], Michael W. Fouts[2], Ali Baheri[2¤], David S. Mebane[1,2]*

**1** National Energy Technology Laboratory, Morgantown, WV, United States of America, **2** Department of Mechanical and Aerospace Engineering, West Virginia University, Morgantown, WV, United States of America

¤ Current address: Department of Mechanical Engineering, Rochester Institute of Technology, Rochester, NY, United States of America
* david.mebane@mail.wvu.edu

**Data Availability Statement:** All relevant data are within the manuscript and its Supporting information files.

**Funding:** Award 1: DSM and KH, U.S. Department of Energy / National Energy Technology Laboratory

## Abstract

A promising approach for scalable Gaussian processes (GPs) is the Karhunen-Loève (KL) decomposition, in which the GP kernel is represented by a set of basis functions which are the eigenfunctions of the kernel operator. Such decomposed kernels have the potential to be very fast, and do not depend on the selection of a reduced set of inducing points. However KL decompositions lead to high dimensionality, and variable selection thus becomes paramount. This paper reports a new method of forward variable selection, enabled by the ordered nature of the basis functions in the KL expansion of the Bayesian Smoothing Spline ANOVA kernel (BSS-ANOVA), coupled with fast Gibbs sampling in a fully Bayesian approach. It quickly and effectively limits the number of terms, yielding a method with competitive accuracies, training and inference times for tabular datasets of low feature set dimensionality. Theoretical computational complexities are $\mathcal{O}(NP^2)$ in training and $\mathcal{O}(P)$ per point in inference, where $N$ is the number of instances and $P$ the number of expansion terms. The inference speed and accuracy makes the method especially useful for dynamic systems identification, by modeling the dynamics in the tangent space as a static problem, then integrating the learned dynamics using a high-order scheme. The methods are demonstrated on two dynamic datasets: a 'Susceptible, Infected, Recovered' (SIR) toy problem, along with the experimental 'Cascaded Tanks' benchmark dataset. Comparisons on the static prediction of time derivatives are made with a random forest (RF), a residual neural network (ResNet), and the Orthogonal Additive Kernel (OAK) inducing points scalable GP, while for the timeseries prediction comparisons are made with LSTM and GRU recurrent neural networks (RNNs) along with the SINDy package.

site support contract (Leidos), Subcontract No. P010220883 Task 30. http://netl.doe.gov. The funders approved the decision to publish and the overall study design, data collection and analysis, and the final manuscript. Award 2: DSM and MWF, U.S. National Science Foundation award 2119688. The funders had no role in study design, data collection and analysis, decision to publish, or preparation of the manuscript. Disclaimer: This project was funded by the United States Department of Energy, National Energy Technology Laboratory, in part, through a site support contract. Neither the United States Government nor any agency thereof, nor any of their employees, nor the support contractor, nor any of their employees, makes any warranty, express or implied, or assumes any legal liability or responsibility for the accuracy, completeness, or usefulness of any information, apparatus, product, or process disclosed, or represents that its use would not infringe privately owned rights. Reference herein to any specific commercial product, process, or service by trade name, trademark, manufacturer, or otherwise does not necessarily constitute or imply its endorsement, recommendation, or favoring by the United States Government or any agency thereof. The views and opinions of authors expressed herein do not necessarily state or reflect those of the United States Government or any agency thereof.

**Competing interests:** The authors have declared that no competing interests exist.

# Introduction

Gaussian processes (GPs) are stochastic functions that are engines for nonparametric regression. Initially developed for modeling and interpolation in geographic information systems datasets, applications have multiplied across many fields of data science. A key advantage of the GP is its broad, continuous nonparametric support and the amenability of different GP kernels to precise analysis. They are widely recognized as powerful vehicles for static dataset modeling and interpolation with uncertainty quantification.

A GP is Gaussian in that it is a covariance model linking pairs of points on functional draws. As such a GP is completely described by a mean function (often zero in the prior) and covariance kernel. The most famous and perhaps simplest of the covariance kernels is the squared exponential:

$$\kappa(x, x') = \varsigma^2 \exp\left[-\frac{(x - x')^2}{\xi}\right] \tag{1}$$

where the sill $\varsigma^2$ and range $\xi$ parameters determine the scale and smoothness of the draws. In a typical implementation modeling a static dataset $Z$, the statistical model

$$Z = \delta(\mathbf{x}|\varsigma^2, \xi) + \epsilon \tag{2}$$

with $\delta \sim \mathcal{N}(0, \kappa)$ the GP and $\epsilon$ an observation error process, is first used to infer the hyperparameters, after which predictions conditioned on the training dataset can be made. The draws on the squared exponential GP—a limiting case of the Matérn covariance family—are infinitely differentiable.

From a practical standpoint the training of the above GP is $\mathcal{O}(N^3)$, where $N$ is the number of training data points, requiring a Cholesky decomposition of the full covariance matrix. This limits the use of the GP to moderately-sized datasets, generally of a thousand instances or fewer. An important avenue of research is accelerating the speed of both training and inference, such that the GP's superior modeling features can be trained on large datasets and deployed as machine learning vehicles within other types of models.

## Scalable Gaussian processes with inducing points

Liu, *et al.* [1] provide a thorough overview of efforts that aim to improve scalability while maintaining prediction accuracy using global kernel approximations derived in some sense from a set of $M << N$ inducing points [2–6]. Generally the goal is to approximate the full-rank kernel matrix with local approximations. Of particular note is a $\mathcal{O}(N)$ method that directly estimates the covariance with training and inference times that limits the increase in $M$ for large $N$ developed by Wilson, *et al.* [7]. Some methods employ Analysis of Variance (ANOVA) decompositions to the full kernel which break out contributions in terms of features and their combinations:

$$\kappa(\mathbf{x}, \mathbf{x}') = \sum_{i=1}^{n} \kappa_i(x_i, x'_i) + \sum_{i=1}^{n-1} \sum_{j=i+1}^{n} \kappa_i(x_i, x'_i)\kappa_j(x_j, x'_j) + \cdots \tag{3}$$

where $n$ is the feature set dimensionality and the individual terms are not necessarily orthogonal. This presents opportunities for variable selection [8]; of particular note is the recent Orthogonal Additive Kernel (OAK) which orthogonalizes the kernels in (3) in order to minimize overlap between main effects and higher-order interactions [9].

Inducing points acceleration of GPs opens up GP regression to large datasets. However, inference is still $\mathcal{O}(M^2)$ per point, limiting its usefulness in contexts where inference speed is

important, such as those where the GP model will be used in the context of control or optimization applications. This motivates a search for alternative methods that are fast in both training and evaluation.

## Karhunen-Loève decomposition and BSS-ANOVA

Another approach to scalability in GPs that is distictive to the inducing points approach is the Karhunen-Loève (KL) expansion, in which the kernel is expressed in terms of a sum over its eigenfunctions:

$$\delta(x; \boldsymbol{\beta}) = \sum_{i=1}^{P} \beta_i \phi_i(x) \tag{4}$$

where

$$\phi_i(x) = \sqrt{\lambda_i} u_i(x)$$

$$\int \kappa(x, x') u_i(x') dx' = \lambda_i u(x)$$

$$\beta_i \sim \mathcal{N}(0, 1)$$

Such methods have the potential to be fast: $\mathcal{O}(NP^2)$ in training and $\mathcal{O}(P)$ per point for inference, where $P$ is the number of terms in the expansion. However such kernels have not been the subject of much research in machine learning contexts generally. The main issues are tractable calculation of the basis functions $\{\phi_i\}$ and a curse of dimensionality [10].

In 2009 Reich and collaborators [11] introduced the Bayesian Smoothing Spline ANOVA (BSS-ANOVA) kernel, which is subject first to an ANOVA decomposition, followed by a KL decomposition. The core of the BSS-ANOVA kernel is:

$$\kappa_1(x, x') = \mathcal{B}_1(x)\mathcal{B}_1(x') + \mathcal{B}_2(x)\mathcal{B}_2(x') - \frac{1}{24}\mathcal{B}_4(|x - x'|) \tag{5}$$

where $\mathcal{B}_k$ is the $k^{\text{th}}$ Bernoulli polynomial, defined by the generating function

$$\frac{te^{tx}}{e^t - 1} = \sum_{i=0}^{\infty} \mathcal{B}_i(x) \frac{t^i}{i!} \tag{6}$$

yielding

$$\mathcal{B}_1(x) \quad = x - \frac{1}{2}$$

$$\mathcal{B}_2(x) \quad = x^2 - x + \frac{1}{6}$$

$$\mathcal{B}_4(x) \quad = x^4 - 2x^3 + x^2 - \frac{1}{30}$$

This kernel is effectively a sum of a non-stationary quadratic response surface—corresponding to the first two terms in (5)—and a stationary deviation (the final term). Covariances for higher-order interactions are constructed with dyadic products of the main effect covariance given in (3):

$$\kappa_2([x_j, x_k], [x'_j, x'_k]) = \kappa_1(x_j, x'_j)\kappa_1(x_k, x'_k) \tag{7}$$

and so on for higher-order interactions. Terms are then multiplied by scaling hyperparameters

and added together to produce the full kernel:

$$\kappa = \sigma_0^2 \tau_0^2 + \sigma_1^2 \tau_1^2 \sum_{i=1}^{n} \kappa_{1,i} + \sigma_2^2 \tau_2^2 \sum_{i=1}^{n-1} \sum_{j=i+1}^{n} \kappa_{2,ij} + \cdots \tag{8}$$

The kernel so constructed is supported on a second-order Sobolev space [11], a broad support which is one of its primary advantages. The significance of separate $\sigma^2$ and $\tau^2$ scaling parameters will become clear below in the context of Bayesian linear regression.

Building the kernel in this fashion effectively addresses the problem of generating the eigenfunctions from the KL decomposition: because all of the terms in (8) are based on the generative kernel (5), the KL decomposition of (8) will depend only on eigenfunctions of $\kappa_1$. Additionally if all input features are normalized to an [0, 1] interval (we restrict the discussion to continuous input features for now), then it is only necessary to compute a single set of basis functions $\{\phi_i\}$. The decomposed BSS-ANOVA GP is written:

$$\delta(\mathbf{x}; \boldsymbol{\beta}) = \beta_0 + \sum_{i=1}^{n} \sum_{k=1}^{\infty} \beta_{ik} \phi_k(x_i) + \sum_{i=1}^{n-1} \sum_{j=i+1}^{n} \sum_{k=1}^{\infty} \sum_{l=1}^{\infty} \beta_{ik,jl} \phi_k(x_i) \phi_l(x_j) + \cdots \tag{9}$$

Given the assumption

$$\sigma_0^2 \tau_0^2 = \sigma_1^2 \tau_1^2 = \sigma_2^2 \tau_2^2 = \cdots = \sigma^2 \tau^2 \tag{10}$$

then the priors for the coefficients $\boldsymbol{\beta}$ are iid normal

$$\beta_{\cdot,\cdot} \sim \mathcal{N}(0, \sigma^2 \tau^2) \tag{11}$$

where the notation $\cdot$ indicates an arbitrary index. While any kernel is amenable to a KL decomposition and as such to the variable selection routine described below, the form of the BSS-ANOVA kernel as a sum of scaled terms (which separates the basis decomposition from the hyperparameters) and its broad support make it particularly suitable for many applications, and we restrict our analysis to this kernel for the remainder of the article.

Following [11] we generate the set $\{\phi_i\}$ by producing $\kappa_1$ for a dense grid consisting of 500 intervals on [0, 1], eigendecompose and fit to cubic splines. Fig 1 shows the first 6 basis functions. These basis functions are nonparametric, pairwise orthogonal, and ordered: note the increase in frequency and decrease in amplitude as the orders increase.

## Variable selection

It's clear from (9) that the number of terms in the expansion can increase rapidly, even for low-dimensional input spaces. A key component of applying the GP to a modeling problem is

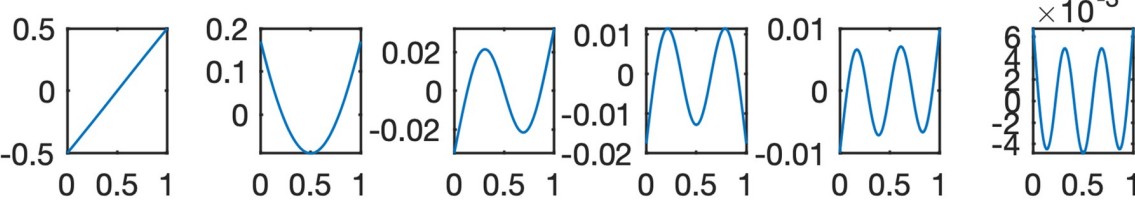

**Fig 1. The first six basis functions of the KL-decomposed BSS-ANOVA kernel.** The basis is nonparametric, spectral, pairwise orthogonal and ordered.

thus the selection of terms. Effectively we seek to minimize the objective function

$$\mathbf{\Phi}(\boldsymbol{\beta}) = ||Z - \delta(\mathbf{x}; \boldsymbol{\beta})||^2 + \zeta(\boldsymbol{\beta}) \tag{12}$$

where $\zeta$ is a penalty function which leads to a sufficiently sparse solution.

## Indicator variable methods

Reich, *et al.* [11] took a hierarchical Bayesian approach to the problem, estimating a separate variance $\tau^2$ for each term in the expansion, which is in turn expressed in terms of an indicator variable with a Bernoulli prior. This approach, like other 'indicator variable' methods, accomplishes the variable selection and the training simultaneously and comprehensively, at the cost of requiring a large number of variables in the prior model and a computationally onerous Markov chain Monte Carlo (MCMC) sampling procedure.

Other sparse optimization methods such as ridge regression or LASSO share the limitation that many high-order terms must be included in the initial model before downselection occurs.

## Forward variable selection

The ordered and orthogonal nature of the basis functions suggests a forward variable selection approach. Rewriting the model (9) for a basis function set of maximum order $q$,

$$\delta(\mathbf{x}; \boldsymbol{\beta}) = \beta_0 + \sum_{i=1}^{n} \sum_{k=1}^{q} \beta_{ik} \phi_k(x_i) + \sum_{i=1}^{n-1} \sum_{j=i+1}^{n} \sum_{k=1}^{q} \sum_{l=1}^{q} \beta_{ik,jl} \phi_k(x_i) \phi_l(x_j) + \cdots \tag{13}$$

then considering a model building procedure which increases $q$ stepwise starting with $q = 1$ reveals that each subsequent step adds $n$ main effect terms (each depending on a single input —$n$ is the number of inputs), $\binom{n}{2} [2(q-1)+1]$ two-way interactions, and $\binom{n}{3} [3(q-1)^2 + 3(q-1)+1]$ three-way interactions. As the model order increases the $L^2$ truncation error for the full kernel decreases as (for the case of a single input) [10]:

$$||\kappa(x, x') - \sum_{i=1}^{q} \phi_i(x) \phi_i(x')|| < \left( \sum_{i=q+1}^{\infty} \lambda_i^2 \right)^{1/2} \tag{14}$$

Since the eigenvalues of the BSS-ANOVA kernel decomposition decrease quickly with increasing order, an approach to the optimization problem (12) focusing on low-order models will sacrifice little in the way of accuracy while realizing significant advantages in both training and inference times.

The design and implementation of such an approach is the main contribution of this work. It approaches the optimization of (12) with an iterative process, finding the most efficient truncation of the system while evaluating the cost function only for candidate models with *fewer* terms than the optimum truncation. The method is fully Bayesian, with a fast linear Bayes sampling procedure at its core. As such the form of the cost function is also Bayesian in nature, taking the form of the Bayesian or Akaike information criteria (BIC/AIC), which incorporate $L^0$

penalties:

$$BIC = P \ln N - 2 \ln \hat{\mathcal{L}} \tag{15}$$

$$AIC = 2P - 2 \ln \hat{\mathcal{L}} \tag{16}$$

where $\hat{\mathcal{L}}$ is the maximum of the likelihood function on the parameter space.

The following sections describe the components of the optimizer in detail and present the implemented variable selection algorithm.

**Linear Bayesian regression.** Given a statistical model with a fixed number of terms

$$z_i = \delta_i(\mathbf{x}_i; \boldsymbol{\beta}) + \epsilon(\sigma^2) \tag{17}$$

with $\epsilon$ a white noise observation error of variance $\sigma^2$ and $\delta$ is a BSS-ANOVA model corresponding to a given truncation to the KL expansion (9), the model is linear in the coefficients $\beta$. Using priors that are conjugate to the likelihood for the coefficients and the observation error variance, a Gibbs sampling methodology can quickly obtain the posterior for the parameters.

The conjugate prior for $\beta$ is zero-mean iid normal: $\beta \sim \mathcal{N}(0, \sigma^2\tau^2 I)$ The conjugate prior for $\sigma^2$ is inverse gamma: $\sigma^2 \sim IG(a, b)$, with $a$ and $b$ the shape and scale parameters, respectively; likewise the conjugate prior for $\tau^2$ is inverse gamma: $\tau^2 \sim IG(a_\tau, b_\tau)$. The Gibbs sampler functions iteratively, such that for fixed $\{\sigma^2, \tau^2\}$, $\beta \sim \mathcal{N}(\mu, \Sigma)$, with

$$\mu = (X^T X + 1/\tau^2 I)^{-1} X^T Z \tag{18}$$

$$\Sigma = \sigma^2 \left( X^T X + 1/\tau^2 I \right)^{-1} \tag{19}$$

where $X \in \mathbb{R}^{N \times P}$ is a matrix constructed from the basis functions appearing in the expansion. Its rows correspond to instances and columns to terms in the expansion. For fixed $\{\beta, \tau^2\}$, $\sigma^2 \sim IG(a^*, b^*)$, with

$$a^* = a + N/2 + P/2 \tag{20}$$

$$b^* = b + \frac{1}{2} \left[ (\mu - \beta)^T (X^T X + 1/\tau^2 I)(\mu - \beta) + Z^T Z - \mu^T X^T Z \right] \tag{21}$$

For fixed $\{\beta, \sigma^2\}$, $\tau^2 \sim IG(a_\tau^*, b_\tau^*)$, with

$$a_\tau^* = a_\tau + P/2 \tag{22}$$

$$b_\tau^* = b_\tau + \frac{1}{2\sigma^2} \beta^T \beta \tag{23}$$

For more details see [12].

**Optimization via forward search.** The algorithm constructs models with terms having up to three-way interactions. The algorithm runs sequentially through stages labeled by an integer index $\mathcal{I}$ initialized at 1. At each stage, a set of trial terms come in to the expansion for which the sum over all basis functions appearing in the term add up to $\mathcal{I}$. Since there are many permutations of basis functions that form this sum, and for any given set of function orders in a term, different ways they can be permuted among the inputs, adding terms at each stage occurs in substages. Each substage adds terms corresponding to a particular set of basis functions, and includes all input permutations. For example: stage 1 adds only first order main

effects: $\phi_1(x_1)$ and $\phi_1(x_2)$ for a two-input dataset. Stage 2 adds second order main effects and first order two way interactions—$\phi_2(x_1)$, $\phi_2(x_2)$ and $\phi_1(x_1)\phi_1(x_2)$—in two substages. The substages occur such that terms involving lower-order basis functions come first. For the example in the case of stage 2, this is the first order two-way interactions $\phi_1(x_1)\phi_1(x_2)$. Each substage adds at once all combinations of inputs and all permutations among each combination, such that each substage adds $2\binom{n}{2}$ terms for two-way interactions and $6\binom{n}{3}$ terms for three-way interactions (for the case where all function orders are different; fewer when two or more are the same). With trial terms added at a given substage, the sampler estimates model parameters and calculates the BIC or AIC. The routine terminates at an optimum BIC or AIC. Because there can be local minima, a "tolerance" setting controls how many substages the algorithm can iterate through without finding a new minimum BIC or AIC before it terminates. The terminated algorithm returns the optimum model.

**Algorithm 1** BSS-ANOVA forward variable selection algorithm

```
1:  procedure FwdVarSelect(x, z, φ, tol)
2:        ▷ x is a matrix of inputs, columns are features and rows are
             instances
3:                                  ▷ z is a column vector of data
4:                            ▷ φ is an ordered set of basis functions
5:                                  ▷ tol is an integer tolerance
6:   Form a column vector of ones ℝᴺ ∋ X = 1
7:   Set 𝓘 = 1
8:   Set count = 0
9:   while count < tol do
10:      Set 𝓩 = {j ∈ ℤ : 0 ≤ j ≤ 𝓘}
11:      Set 𝓠 = {(i ∈ 𝓩, j ∈ 𝓩, k ∈ 𝓩) : i + j + k = 𝓘}
12:      Order q ∈ 𝓠 s.t. qₘ ≺ qₙ when max(qₘ) < max(qₙ)
13:      for l = {1, 2, ⋯ |𝓠|} do
14:         Set m_d = {q₁, 0, 0, ⋯} s.t. |m_d| = |x|
15:         Form 𝓜_d = S(m_d)        ▷ 𝓜_d contains all permutations of m_d
16:                    ▷ Each element of 𝓜_d is a term in the expansion
17:         Build X_d where X_{d,ij} = ∏_{k:𝓜_{d,jk}≠0} φ_{𝓜_{d,jk}}(x_{ik})
18:         Recursively concatenate: X = [XX_d], 𝓜 = [𝓜; 𝓜_d]
19:         Call the sampler: β, BIC = gibbs(X, z, φ)
20:         if the BIC is a minimum for all models then
21:            count = 0
22:         else
23:            count = count + 1
24:      𝓘 = 𝓘 + 1
25:   Return 𝓜, β, BIC
```

## Experiments: Dynamic system identification

### Procedure

BSS-ANOVA regression—as is the case for other GPs—is most effective for tabular datasets with continuous inputs and targets of moderate dimensionality. This suggests an application in dynamic systems identification. Indeed BSS-ANOVA GPs have been utilized as components of other models ("intrusively") for this purpose in a number of applications [13–15]. We demonstrate here that they may also be used directly to identify dynamics in more general cases, without the aid of an accompanying model. In treating the dynamic systems application we should be clear to make a distinction between the present approach in which the GP models the static relationship between the time derivatives of states and the states and forcing functions themselves, and other "dynamic GP" applications, *e.g.* [16].

The procedure is a concurrent one, in that time derivatives estimated from the datasets are modeled directly using BSS-ANOVA with forward variable selection, using the concurrent values of the system states and other inputs; for example a two-state system is modeled using two separate GPs:

$$\dot{x}_1 = \delta_1(x_1, x_2, u) \tag{24}$$

$$\dot{x}_2 = \delta_2(x_1, x_2, u) \tag{25}$$

The identified system is then integrated to yield predictions with uncertainty.

The procedure was demonstrated on two nonlinear dynamic datasets: a synthetic dataset derived from the susceptible, infected, recovered model (SIR model) for infectious disease, and the 'Cascaded Tanks' experimental benchmark dataset. In both cases comparisons were made to long short term memory (LSTM) and gated recurrent unit (GRU) neural networks, along with the sparse identification of nonlinear dynamical systems (SINDy) package [17] for time-series prediction. In the case of the cascaded tanks benchmark comparisons were made against random forest (RF), a residual neural network (ResNet) and the state-of-the-art OAK inducing points scalable GP [9] for the static derivative estimation problem.

## Experimental benchmark: Cascaded tanks

The cascaded tanks nonlinear benchmark dataset is an experimental nonlinear dynamic system [18]. The experiment consists of a set of two tanks and a reservoir of water. An upper tank is filled by a pump from the reservoir. An outlet in the upper tank empties into the lower tank, which in turn empties through an outlet back into the reservoir. A signal sent to the pump serves as the forcing function for the system, with the tank water level heights the two states of the system.

Hyperparameters are an important component of any comparison of numerical methods. While sometimes it is advantageous to perform hyperparameter optimization via sweeps or other methods, it is also important to compare methods using reasonable values that are in the neighborhood of commonly used defaults, with trial-and-error adjustments, as this is how most users will approach the task. This is therefore the approach we took in the following comparisons. An exception is for the comparisons with the SINDy package, for which a sweep was performed as described below.

We first compared the performance of BSS-ANOVA with RF, ResNet and OAK static regressors. Derivatives were calculated via direct finite differences for the relatively noise-free dataset, yielding 10000 instances. Each method was trained on concurrent values of both states and the forcing function for each derivative. For the GP we used hyperparameters of $a = 1000$, $b = 1.001$, $a_\tau = 4$ and $b_\tau = 55$ for $\dot{h}_1$ and 69.1 for $\dot{h}_2$, with tolerances of 3 for $\dot{h}_1$ and 5 for $\dot{h}_2$, and the AIC as discriminator. Inputs were normalized to [0, 1] using min-max scaling. Of 2000 draws the first 1000 were discarded. Only two-way interactions were required. For the RF 100 trees were used with a leaf size of 5. The ResNet had a depth of 6 (filter sizes ranging from 16 to 64) and in between each fully connected layer is a batch normalization and relu layer. The mini batch size is 16, initial learn rate is 0.001, the data was shuffled every epoch for a total of 30 epochs, and the validation frequency was 1000. OAK was applied at a maximum dimension of 3 and with the default value of 200 inducing points. The 5-fold cross-validated results appear in Table 1. OAK performed best for both outputs, followed closely by BSS-ANOVA. Both GPs outperformed the RF and the ResNet by clear margins.

The nature of the decomposed GP as a sum over terms pertaining explicitly to certain inputs is a distinct advantage of the KL decomposition. The results indicate that the most

**Table 1. Cascaded tanks 5-fold cross validated accuracies: Derivatives, mean absolute error (MAE).**

| Method | $\dot{h}_1$ (MAE/$10^{-4}$) | $\dot{h}_2$ (MAE/$10^{-4}$) |
|---|---|---|
| OAK | 17±4.7 | 36±2.4 |
| BSS-ANOVA | 18±6.5 | 39±3.6 |
| ResNet | 36±14 | 61±15 |
| RF | 30±9.4 | 49±4.9 |

important factors in the timeseries model for tank 1 are the water levels in both tanks, with tank 2 slightly more important. This is counter-intuitive, since the leve in tank 2 (the lower tank) does not even appear in a naive physical model of the system [18]. The analysis shows that the actual system contains significant feedback. For the dynamics of the level in tank 2, the most important term is an interaction between the tank 2 level (second order basis function) and the pump signal (first order).

Timeseries predictions follow for the GP via a 4$^{\text{th}}$-order Runge-Kutta integration routine. These were compared with LSTM and GRU recurrent neural networks (RNNs), along with the SINDy package. For the LSTM there was one LSTM layer and a total of 128 hidden layers, the data was shuffled every epoch for a maximum of 125 epochs, verbose was equal to 0, and the sequence was padded to the left. The GRU had one GRU layer and 150 total hidden layers, the data was shuffled every epoch for a total of 150 epochs, verbose was equal to zero and the sequence was padded to the left. The 5-fold cross-validated results (datapoints were not randomized before creating the folds so as to preserve the timeseries order) appear in Table 2. BSS-ANOVA is most accurate, followed by the LSTM and the GRU. Fig 2 shows the predictions of the GP and the LSTM for the upper tank for one of the test folds. The GP predictions are superior near the sharp inflection and critical points where nonlinearities are strongest. Note that the first 50 points of each test set, which were provided to the LSTM and GRU as a start-up set in the prediction phase, were removed from the calculation of error for both methods.

To evaluate the SINDy performance on the Cascading Tanks task, an exploratory analysis was performed over the hyperparameters (optimizer, threshold, alpha, and basis function libraries). Optimizers were first tested using default alpha (0.05) and basis functions (2nd order polynomials), and a threshold of 0.001 due to the small coefficients of the model terms. Of the 8 optimizers tested, 2 produced errors and were not able to be evaluated. Of the 6 remaining optimizers, all produced similar results. STLSQ, SR3, and Constrained SR3 were marginally the best performing models and were used for basis function evaluation.

1st, 2nd, 3rd, and 5th order polynomials, and 3rd order polynomials with the addition of Fourier functions were tested successfully. The largest improvement was seen moving to 3rd order polynomials with an additional small improvement adding in the Fourier functions. 4th

**Table 2. Cascaded tanks 5-fold cross validated accuracies: Timeseries, mean absolute error (MAE).**

| Method | $h_1$ (MAE) | $h_2$ (MAE) |
|---|---|---|
| BSS-ANOVA | 0.1167±0.0382 | 0.1577±0.0334 |
| SINDy | 0.1391±0.0631 | 0.1768±0.0695 |
| LSTM | 0.2345±0.1006 | 0.2296±0.0378 |
| GRU | 0.3243±0.1092 | 0.2481±0.0402 |

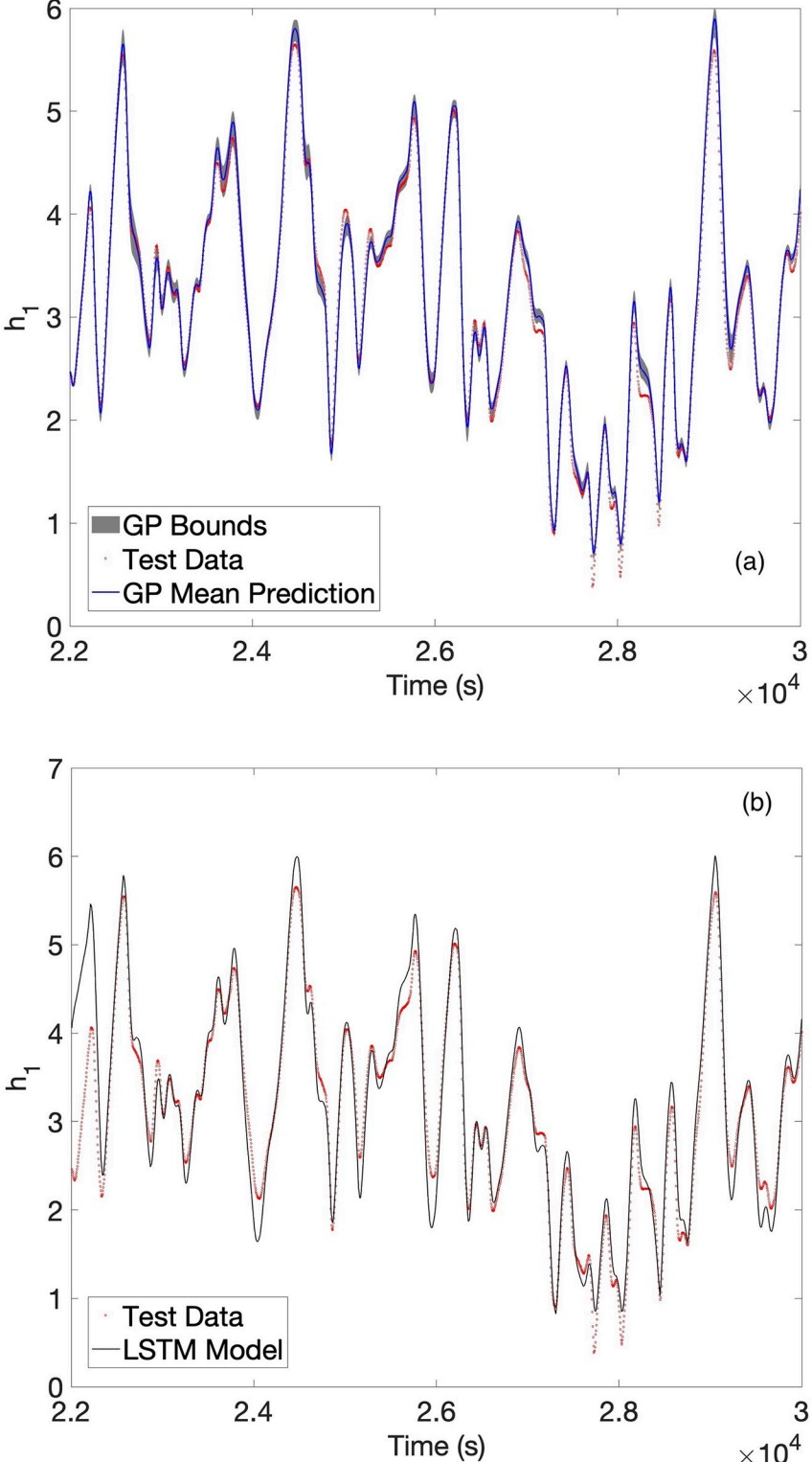

**Fig 2.** (a) BSS-ANOVA and (b) LSTM predictions vs. test set data for the water level height in tank 1 of the cascaded tanks dataset. Shaded regions in (a) are 95% confidence bounds estimated from a draw of 40 curves.

order polynomials and 2nd/4th order polynomials with Fourier functions all failed to converge.

The three chosen optimizers were used with 3rd degree polynomials with Fourier functions in a threshold parameter search to find the best results with a default alpha of 0.05. The optimum occurred at a threshold of 0.0001 using the STLSQ optimizer. Lastly, a search was performed over alpha to determine the optimal value which occurred at 0.05 (default).

While it is reasonable to expect that OAK with 200 inducing points would outperform BSS-ANOVA in the time integration, it was not practical to make this comparison for reasons of computing time. A comparison with a reduced number of inducing points and increased time step in the integrator was made—results are discussed below.

Comparing the results of the sparse KL-decomposed BSS-ANOVA method with other published treatments of the same benchmark is difficult because of the lack of rigorous separation between training and test data in most applications. One study of note is that of La Cava, et al. [19], who used an approach to reconstructing the dynamic system from a predetermined set of parametric terms using a version of genetic programming. Assuming clear separation in this study between test and train sets, the approach described in this paper outperforms the "Epigenetic Linear Genetic Programming" approach by over an order of magnitude in mean square error, while also outperforming the NARX-NN approach (a lightly parameterized recurrent neural network) offered as comparison in [19] by approximately one order of magnitude.

## Synthetic benchmark: Susceptible, infected, recovered model

The susceptible, infected, recovered model (SIR model) is a common simulation for infectious disease. Though there are several versions, the simplest is three states, only two of which are independent. The system is written

$$\dot{S} = -BIS/N_P \tag{26}$$

$$\dot{I} = BIS/N_P - \gamma I \tag{27}$$

$$\dot{R} = \gamma I \tag{28}$$

where $S(t)$ is the susceptible population, $I(t)$ the infected, $R(t)$ the recovered, $B(t)$ is the transmissibility (which we utilize as a forcing function), $\gamma$ is the recovery rate (which we leave fixed at 0.5) and $N_P$ is the total population. Because $N_P$ is fixed and $S + I + R = N_P$, only two states are independent, so the system dynamics can be captured by modeling only two of the three. We chose $I(t)$ and $R(t)$.

The training data consists of 58 curves. All curves in the training set have a fixed $B$ value ranging from 0.5 to 9, in six intervals of 1.7. For each value of $B$ there are 8–10 siumulations corresponding to different initial conditions designed in such a way to provide coverage of the state space. (Exact initial conditions used appear in the supplement.) Each simulation used $N_P$ = 1000.

The test data consists of 24 curves, each of which features a temporally changing transmissibility $B(t)$. There are three initial $B_0$ values: 1.35, 4.75 and 8.15. For each starting point there are two types of transmissibility curves: a ramp and a sinusoid. The $B_0 = 1.35$ and $B_0 = 4.75$ starting points have ramps with a positive slope of 1, while the $B_0 = 8.15$ curves have a slope of -1. All ramps run from $t = 0$ to $t = 4$, where they level off. The sinusoids have amplitudes between 0.5 and 3 and a period of 1.

Hyperparameters for BSS-ANOVA were: $a = a_\tau = 4$ for both states, $b_{\tau,R} = 8.95$ and $b_{\tau,I} = 72.1$, while $b_I = 1.25$ and $b_R = 20$. 2000 draws were taken and the first 1000 discarded. The

**Table 3. SIR test set results for BSS-ANOVA and SINDy: Mean absolute error (MAE).**

| Method | $I$ (MAE) | $R$ (MAE) |
|---|---|---|
| BSS-ANOVA | 5.2739±4.0183 | 11.8345±21.7337 |
| SINDy | 1.8722±4.1736 | 8.0659±22.2364 |

tolerance was 6. Hyperparameters for SINDy, LSTM and GRU were the same as for the Cascaded Tanks.

Results for BSS-ANOVA and SINDy are shown in Table 3. SINDy performed better (in average) on both state predictions. Statistics were not calculated for the GRU and LSTM as each failed to replicate the dynamics in most test cases and were obviously inferior to both BSS-ANOVA and SINDy in every instance. A graphical comparison for a selected number of curves from the test set are shown in Fig 3. The largest overall term in the model for the infected population was a two-way interaction between the infected state and recovered state, consisting of second and third order basis functions.

## Training and inference times

Training and inference times for BSS-ANOVA were fast, with a mean total train time of 6.3 seconds for the cascaded tanks and 10.8 seconds for the SIR, with 8,000 and 20,000 training data points, respectively, on a 2019 6-core i7 processor with 16 GB of RAM. The routines were implemented in MATLAB, but not parallelized or optimized for speed. Models for $\dot{h}_1$ contain between 23 and 41 terms, while $\dot{h}_2$ has between 38 and 57 terms. Prediction times for 2000 static points for the cascaded tanks averages 0.5437 s, and the time for evaluating integrals over the test set averages 20.22 s. For the SIR model the $\dot{I}$ model had 81 terms and the $\dot{R}$ model 9 terms, with a mean integration time of 5.3 s. Analyses have shown that the rate limiting step in BSS-ANOVA build algorithms are the $\mathcal{O}(NP)$ construction of the $X$ matrix from the inputs and basis functions. The neural networks were native MATLAB functions, parallelized and optimized for speed. Nonetheless train times were considerably longer, with mean train times of 130s for the ResNet and 175 and 123 s, respectively, for training the LSTM and GRU for the cascaded tanks. This is to be expected given that the number of weights in the neural nets are on the order of $10^4$.

It was not feasible to integrate OAK at the level of 200 inducing points to the same standard as that of BSS-ANOVA because of time considerations. A reduced set of 40 inducing points yielded accuracies in the static estimation problem that were approximately the same as BSS-ANOVA. A reduced time step (500 vs. 20,000 integration steps) brought the integration time down to 51 minutes for OAK, with MAE/MAPE of 0.1554/6.3 for $h_1$ and 0.2378/9.1 for $h_2$. Reducing the integration step to the same level as BSS-ANOVA (where we could expect comparable integration accuracies) would require approximately 33 hours.

## Discussion

The results show that the forward variable selection methodology makes the KL-decomposed GP a viable option for dynamic systems—competitive in these preliminary results with state-of-the-art routines in both static and dynamic modeling tasks—due to its combination of speed and accuracy. In addition the Bayesian nature of the method yields estimates of uncertainty in the predictions, which can be useful in design of experiments and optimization. It also creates opportunities for fast Bayesian model updates in a control context, wherein only

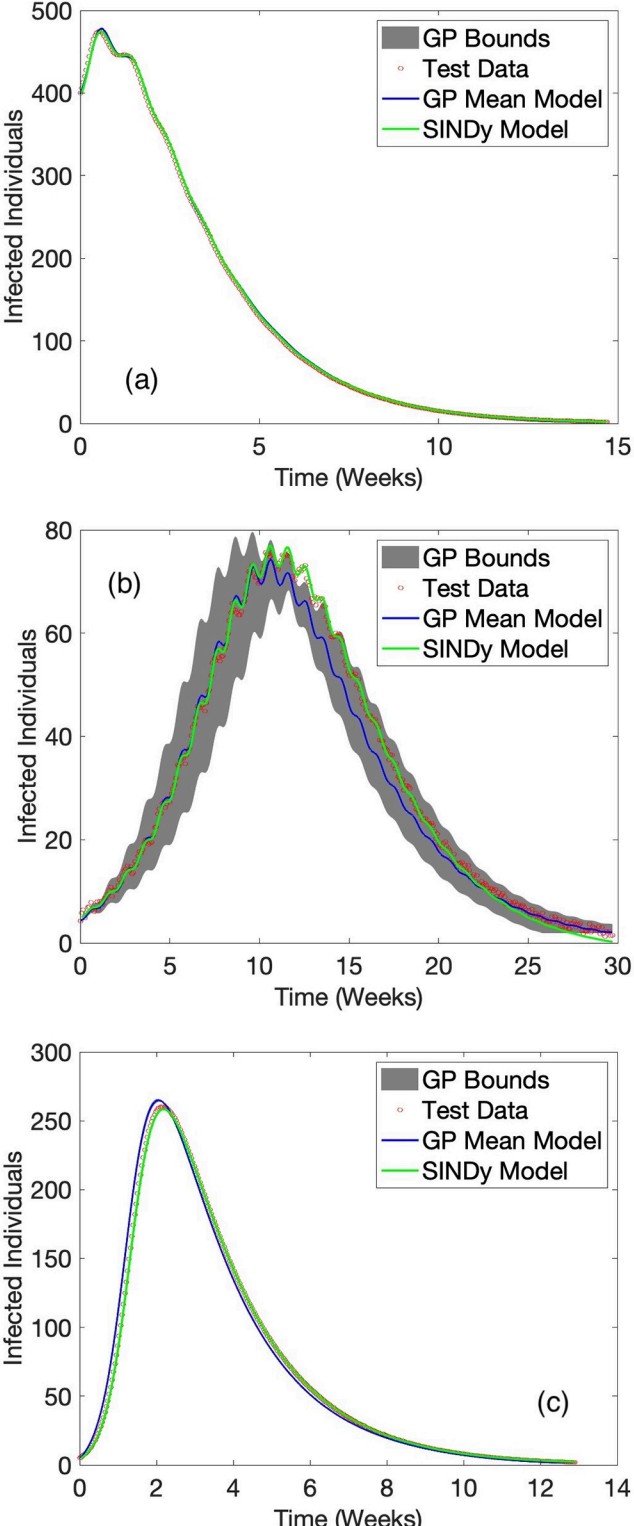

**Fig 3. BSS-ANOVA and SINDy predicted results for *I* in three representative test datasets.** (a) and (b) have sinusoidal and and (c) upward ramp control function dynamics that were not present in the training set. Red circles are test data, the blue curve is the mean and grey shading the 95% confidence bounds for BSS-ANOVA, while the green curve is SINDy with 3rd-order polynomials plus Fourier terms.

new data need be taken into account due to the presence of a strong prior probability distribution constructed from previously-utilized data.

The neural networks were not able to obtain the dynamics in the SIR test because the RNNs map the recurrent inputs to outputs directly on the Hilbert space of time-dependent states and control functions. That is, if the specific type of time-dependent control function behavior found in the test set does not appear in the training set—as is the case in the SIR task—then the test set is out-of-distribution for the RNN. SINDy and BSS-ANOVA by contrast construct static models of the system in the (much more tractable) Euclidean state/control function space. Since the test set forcing function never exceeds the bounds of the training set in that Euclidean space, the test set is in-distribution for both methods.

More speculatively, the performance difference between SINDy and BSS-ANOVA on the different tasks may arise from the relative suitability of the basis to the different types of datasets represented. The nonparametric GP basis was better suited to the experimental benchmark (where the dynamics are nonparametric) while the $3^{\text{rd}}$-order polynomial SINDy basis excelled in the synthetic case that arises from dynamics which are in fact generated from first, second and third-order polynomials. It will be interesting to explore this hypothesis further in future experiments.

There are several ways in which future work might improve on the current algorithm. A more discerning selection of terms at each substage in the variable selection routine—adding some, but not necessarily all terms at each stage—may improve performance by limiting overfitting while also automatically selecting features. Here we have been exploring the use of simulated annealing and Markov Chain Monte Carlo approaches modified to retain the speed advantages of fast Gibbs sampling as potential innermost optimization processes. Such potential improvements along with code optimization and GPU acceleration will be included in future versions of the method.

## Conclusion

A new forward variable selection algorithm has made the scalable Gaussian process BSS-A-NOVA a fast and accurate method for nonparametric regression of tabular data on continuous input spaces. The speed and accuracy for this type of dataset makes it an advantageous method for dynamic system identification. Favorable comparisons with successful and popular sparse basis and neural network approaches for timeseries problems were made in prediction tasks for a pair of nonlinear synthetic and experimental dynamic datasets.

## Supporting information

**S1 File. Code and data used in the production of this manuscript: MATLAB results for all methods except SINDy.**
(ZIP)

**S2 File. Code and data used in the production of this manuscript: SINDy results for Cascading Tanks.**
(ZIP)

**S3 File. Code and data used in the production of this manuscript: SINDy results for SIR.**
(ZIP)

## Acknowledgments

Many thanks to Charlie Harmison and Josh Caswell for producing some of the neural network results.

## Author Contributions

**Conceptualization:** David S. Mebane.

**Data curation:** Kyle Hayes, Michael W. Fouts.

**Formal analysis:** David S. Mebane.

**Funding acquisition:** David S. Mebane.

**Investigation:** Kyle Hayes, Michael W. Fouts, Ali Baheri, David S. Mebane.

**Methodology:** Ali Baheri, David S. Mebane.

**Project administration:** David S. Mebane.

**Software:** Kyle Hayes, Michael W. Fouts, David S. Mebane.

**Supervision:** David S. Mebane.

**Validation:** Kyle Hayes, Michael W. Fouts.

**Visualization:** Kyle Hayes, Michael W. Fouts.

**Writing – original draft:** Ali Baheri, David S. Mebane.

**Writing – review & editing:** David S. Mebane.

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
