## [Decision Letter · Decision Letter 0]

21 Jul 2023

PONE-D-23-14436Forward variable selection enables fast and accurate dynamic system identification with Karhunen-Loève decomposed Gaussian processesPLOS ONE

Dear Dr. Mebane,

Thank you for submitting your manuscript to PLOS ONE. After careful consideration, we feel that it has merit but does not fully meet PLOS ONE’s publication criteria as it currently stands. Therefore, we invite you to submit a revised version of the manuscript that addresses the points raised during the review process.

Please find the reviews from two reviewers below. Please note that both reviewers have expressed concerns that the algorithm and motivations behind its structure and implementation is not clear, and we ask that you thoroughly address this in order to ensure that the manuscript is fully reproducible.

We look forward to receiving your revised manuscript.

Kind regards,

Hanna Landenmark

Staff Editor

PLOS ONE

“Many thanks to Charlie Harmison and Josh Caswell for producing some of the neural

network results.

DSM and KH were supported by the U.S. Department of Energy, National Energy

Technology Laboratory through a site support contract: Subcontract No. P010220883

Task 30

DSM and MWF were supported by the National Science Foundation under award

2119688”

 “Award 1: DSM and KH, U.S. Department of Energy / National Energy Technology Laboratory site support contract (Leidos), Subcontract No. P010220883 Task 30. http://netl.doe.gov. The funders approved the decision to publish and the overall study design, data collection and analysis, and the final manuscript.

Award 2: DSM and MWF, U.S. National Science Foundation award 2119688. The funders had no role in study design, data collection and analysis, decision to publish, or preparation of the manuscript.”

3. We noted in your submission details that a portion of your manuscript may have been presented or published elsewhere. [There is a preprint copy on arXiv: https://arxiv.org/abs/2205.13676] Please clarify whether this publication was peer-reviewed and formally published. If this work was previously peer-reviewed and published, in the cover letter please provide the reason that this work does not constitute dual publication and should be included in the current manuscript.

4. Please remove your figures from within your manuscript file, leaving only the individual TIFF/EPS image files, uploaded separately. These will be automatically included in the reviewers’ PDF.

Reviewers' comments:

Reviewer's Responses to Questions

**Comments to the Author**

1. Is the manuscript technically sound, and do the data support the conclusions?

Reviewer #1: Partly

Reviewer #2: Partly

2. Has the statistical analysis been performed appropriately and rigorously? 

Reviewer #1: I Don't Know

Reviewer #2: Yes

3. Have the authors made all data underlying the findings in their manuscript fully available?

Reviewer #1: Yes

Reviewer #2: Yes

4. Is the manuscript presented in an intelligible fashion and written in standard English?

Reviewer #1: No

Reviewer #2: Yes

5. Review Comments to the Author

Reviewer #1: This paper focuses on developing scalable Gaussian process methods with applications to accurate dynamic system identification. To do so, the authors adopt the idea of Karhunen-Loeve decomposition and propose a forward variable selection approach to incrementally select the terms in the decomposition. Techniques in Bayes model selection are used to accomplish the goal. The authors provide numerical experiments to showcase the performance of their method.

I found the general idea natural to understand, but the algorithm described in the section "Optimization via forward search" and "Algorithm 1" are VERY DIFFICULT for me to comprehend. The authors should improve the clarity of the writing since the algorithm is the paper's main contribution. For example, can the authors be more explicit about the different stages of the algorithm and how different terms are added? Moreover, can the authors specify how the BIC and AIC are calculated and used? I found these parts poorly explained in the paper, making readers hard to follow what the authors have exactly done.

Other comments:

1. In the explanation of Karhunen-Loeve decomposition on page 4, I am confused about the notations phi_i and u_i. Is u_i normalized? If so, the covariance of (4) seems not equal to k. Please explain. The same question applies to equation (14). Please check.

2. For equations (16)-(21), I would suggest adding a reference or some details so readers can follow more easily.

3. Please be more explicit about the terminology used. For example, the author never explained the meaning of "MAE" (e.g., in Table 1)

4. Notations and grammar:

(1) Page 6, equation (11), beta_{dot, k}. What is the meaning of "dot"?

(2) Page 5, line 59. "supported by" -> "supported on"?

5. The quality of the figure should be improved. Currently, it is hard to read and compare different lines.

Reviewer #2: The authors propose a new approach for variable selection for Gaussian processes by making use of the Bayesian smooting spline ANOVA kernel (BSS-ANOVA) a (numerically determined) Karhunen-Loeve decomposition.

I have a list of comments and suggestions to the authors that should be considered before re-committing the paper:

1. I think it should be pointed out more clearly in the abstract already what exactly is new in the method. What was there before and what exactly is your innovation?

2. I think it should be added in the abstract how efficient the proposed method is, in terms of complexity w.r.t. size of the data set, dimensionality etc.

3. Concerning Eq. (3): Some authors mean by the ANOVA decomposition already a uniquely defined decomposition whichs terms are orthogonal w.r.t. to an inner product. In your setting, the single parts in (3) do not neccessarily have to fulfill some orthogonality condition?

4. A lot of parameters are introduced and re-used some pages later wihtout naming them or refering to them. Reading the paper for the first one will face difficulties. Some notations are a somewhat mixed up and not so clear, example: in Eq. (2) it is delta(x|zeta^^2,xi), then in (4) it is delta(x;beta) ... is it the same?

5. Equation (5): what is the motivation to look at such kernels? What are special properties or advantages in comparison to other frequently applied kernels. Could or should it be used in other frameworks, too?

6. Equation (8): It is not so clear to my, why two types of parameters (taus and sigmas) are needed, they are just multiplied.

7. Equation (12): This function will be minimized with respect to the betas. Which regularization are you coing to be using? Which optimization algorithms and implementations do you use?

8. Equation (14): What are the lambdas?

9. Equations (16) and (17): Using the eigenfunctions and constructing the matrix X, you could switch from the kernel formulation to the primal formulation of a Gaussian process. Is this what is expressed by (16) and (17)? Equation (16) is just like kernel ridge regression or linear regression with non-linear feature map.

10. The Section "Variable Selection" is not clearly structured. What is the relation between the different subsections and sub-subsections (not nicely visible)? What is the final result? Please, refer to the Algorithm below. Complexity is not adressed.

11. In general, I was wondering why sections and subsections are not numbered.

12. ANOVA means analysis of variance. I think, the found betas should give some interpretability (which dimensions are important and which are coupled)!? What insights do you gain from that in the considered numerical examples?

13. Numerical Experiments / Data sets: Please comment a bit on data pre-processing. Your x's have to be scaled to [0,1]. How was that achieved? Already given or via min-max scaling? What is the dimension of the data in the examples? The used hyper parameters have been tuned automatically or tuned/set by hand? What is the outcome of the variable selection procedure? How do the betas look like and which information can you gain from that?

14. Table 1: The results are self-produced. How comparable are the results actually? What are the best results for the data set published in the literature so far?

15. Training and inference times: Tables summarizing and comparing the results would be nice.

16. I think, a lot of research is going on around Gaussian processes at the moment. I was wondering why there is not so much literature cited (only 16 references). There are also recent papers concerning hyperparameter optimization for Gaussian processes (minimizing the negative log-likelihood).

17. I also want to draw your attention to publications by Potts and Schmischke, who make use of ANOVA-decompositions of functions in terms of Fourier partial sums (https://doi.org/10.1137/20M1354921 and follow-up studies) and using it within least-squares regression / Machine Learning applications.

Overall I have the feeling that a very up-to-date toping is discussed with an innovation that is worth beeing published. However, some things should be clarified better, making the paper easier to read and understand. Advantages, strengths, novelty and innovation should be better discussed and highlighted.

I recommend accepting the paper after a revision.

6. PLOS authors have the option to publish the peer review history of their article (what does this mean?). If published, this will include your full peer review and any attached files.

Reviewer #1: No

Reviewer #2: No

---

## [Author Response · Author response to Decision Letter 0]

24 Dec 2023

Dear Hanna Landenmark,

On behalf of the authors, thanks to you and the reviewers for assessing our manuscript. We carefully considered all comments and made substantive changes to the manuscript in response. This document summarizes those changes and provides additional explanations where needed.

1. We are using a PLoS One LaTeX template. Apologies for missing any details on the submission requirements; we have tried to find and follow all of them in the resubmission. (Where possible: we could not find guidance for file naming in the documents appearing at the links you sent or in the general submission guidelines.)

2. Funding information has been removed from the acknowledgements section. The funding information you presented is correct.

3. An unreviewed preprint has been submitted to arXiv.

4. There are no figures in the manuscript file, only placeholders. However we have removed these as well.

Reviewer 1:

Comment: I found the general idea natural to understand, but the algorithm described in the section "Optimization via forward search" and "Algorithm 1" are VERY DIFFICULT for me to comprehend. The authors should improve the clarity of the writing since the algorithm is the paper's main contribution. For example, can the authors be more explicit about the different stages of the algorithm and how different terms are added? Moreover, can the authors specify how the BIC and AIC are calculated and used? I found these parts poorly explained in the paper, making readers hard to follow what the authors have exactly done.

Response: We have attempted to clarify the algorithm using plain English in addition to the more technical and precise symbolic notation. We note that plain descriptions of the algorithm also appear in the text of the manuscript, and the algorithm itself is designed to add precision to those descriptions. There were indeed several aspects of the algorithm that needed clarification and we the revised version is in fact much clearer. We have also reproduced expressions for BIC and AIC – which, as noted in the text, are the objective functions of the optimization.

Comment: In the explanation of Karhunen-Loeve decomposition on page 4, I am confused about the notations phi_i and u_i. Is u_i normalized? If so, the covariance of (4) seems not equal to k. Please explain. The same question applies to equation (14). Please check.

Response: u_i is normalized by the square root of the eigenvalue, a common approach that renders the variances of the coefficients (betas) iid. However there is a mistake here in the specification of the variance of beta. Thanks for catching it – that mistake has been corrected.

Comment: For equations (16)-(21), I would suggest adding a reference or some details so readers can follow more easily.

Response: This is (mostly) textbook Bayesian linear regression. There are many external resources, including Wikipedia. We added one: a blog post written by one of the authors. That post provides more detail and commentary for readers seeking that.

Comment: Please be more explicit about the terminology used. For example, the author never explained the meaning of "MAE" (e.g., in Table 1)

Response: Sorry for the oversight. MAE is “mean absolute error” and we now define that in the table caption. We also added it to subsequent tables.

Comment: Page 6, equation (11), beta_{dot, k}. What is the meaning of "dot"?

Response: In statistics notation it is common to use a dot as a stand-in for an argument or index when it is arbitrary. Here the index i indicates the input, and the equation indicates that the given relationship holds irrespective of input. However it is also true that the variance of the coefficients is independent of the order of the basis function (as was explained in an earlier response) – we corrected this and now have two dots instead of one. A parenthetical explanation of the notation was also given.

Comment: Page 5, line 59. "supported by" -> "supported on"?

Response: Sure.

Comment: The quality of the figure should be improved. Currently, it is hard to read and compare different lines.

Response: Which figure? In any event we have done our best, producing high-resolution graphics. Since the figures are illustrations of the data appearing in the tables we think these are sufficient.

Reviewer 2:

Comment: I think it should be pointed out more clearly in the abstract already what exactly is new in the method. What was there before and what exactly is your innovation?

Response: From the abstract: “This paper reports a new method of forward variable selection, enabled by the ordered nature of the basis functions in the KL expansion of the Bayesian Smoothing Spline ANOVA kernel (BSS-ANOVA), coupled with fast Gibbs sampling in a fully Bayesian approach.” It goes on to describe what was done to demonstrate this and comparisons to methods that have come before. We are always happy to address comments as best we are able, but this one leaves us at a bit of a loss – we are not sure how we could better accomplish what the reviewer is asking for.

Comment: I think it should be added in the abstract how efficient the proposed method is, in terms of complexity w.r.t. size of the data set, dimensionality etc.

Response: Done.

Comment: Concerning Eq. (3): Some authors mean by the ANOVA decomposition already a uniquely defined decomposition whichs terms are orthogonal w.r.t. to an inner product. In your setting, the single parts in (3) do not neccessarily have to fulfill some orthogonality condition?

Response: In our case, it does not. This was made clear in the revision.

Comment: A lot of parameters are introduced and re-used some pages later wihtout naming them or refering to them. Reading the paper for the first one will face difficulties. Some notations are a somewhat mixed up and not so clear, example: in Eq. (2) it is delta(x|zeta^^2,xi), then in (4) it is delta(x;beta) ... is it the same?

Response: Equations (2) and (4) refer to different cases – equation (2) refers to the conventional squared exponential kernel introduced in (1), whereas (4) refers to the decomposed kernel. The only commonality is the use of delta to represent the GP. The new section and the language introducing it should make clear this is a new case.

Comment: Equation (5): what is the motivation to look at such kernels? What are special properties or advantages in comparison to other frequently applied kernels. Could or should it be used in other frameworks, too?

Response: This is a good question. While any kernel is amenable to a KL decomposition, the kernel hyperparameters don’t always come through the decomposition intact in the prior specification for the expansion coefficients – e.g. the range parameter \\xi of the squared exponential kernel given in (1) – as they do for the BSS-ANOVA kernel. There are undoubtedly other kernels that have this property also. However the support of the BSS-ANOVA kernel is a Sobolev space, as noted in the manuscript, which makes it broad enough for many applications. We do expect that different kernels may be better suited to different types of datasets. The variable selection method works for an arbitrary kernel basis. We’ve added related language to the manuscript.

Comment: Equation (8): It is not so clear to my, why two types of parameters (taus and sigmas) are needed, they are just multiplied.

Response: This is common in Bayesian linear regression, where sigma^2 is the variance of the observation error and sigma^2tau^2 is the variance of the coefficients. This becomes clearer in the section on sampling – we’ve added a note near equation 8.

Comment: Equation (12): This function will be minimized with respect to the betas. Which regularization are you coing to be using? Which optimization algorithms and implementations do you use?

Response: Regularization is L0, as embodied by the BIC / AIC, and the algorithm is precisely the topic of the paper: the forward variable selection routine. We have written out the formulas for the BIC and AIC in response to another reviewer’s comment.

Comment: Equation (14): What are the lambdas?

Response: Eigenvalues of the KL decomposition, as defined in (4).

Comment: Equations (16) and (17): Using the eigenfunctions and constructing the matrix X, you could switch from the kernel formulation to the primal formulation of a Gaussian process. Is this what is expressed by (16) and (17)? Equation (16) is just like kernel ridge regression or linear regression with non-linear feature map.

Response: We are not sure what is meant by the ‘primal formulation of a Gaussian process’, but (16) is linear regression for a fixed number of terms. Ridge regression is variable selection with L2 penalties, which does not make use of a forward variable selection method, and would therefore defeat the purpose here.

Comment: The Section "Variable Selection" is not clearly structured. What is the relation between the different subsections and sub-subsections (not nicely visible)? What is the final result? Please, refer to the Algorithm below. Complexity is not addressed.

Response: See above; we modified the algorithm and the corresponding text to try and make it easier to follow.

Comment: In general, I was wondering why sections and subsections are not numbered.

Response: We used the PLoS LaTeX template.

Comment: ANOVA means analysis of variance. I think, the found betas should give some interpretability (which dimensions are important and which are coupled)!? What insights do you gain from that in the considered numerical examples?

Response: This is a good point. We have added a bit more information (beyond what already appears in the experimental section) regarding the regression results that highlights the interpretability of the KL-decomposed GP.

Comment: Numerical Experiments / Data sets: Please comment a bit on data pre-processing. Your x's have to be scaled to [0,1]. How was that achieved? Already given or via min-max scaling? What is the dimension of the data in the examples? The used hyper parameters have been tuned automatically or tuned/set by hand? What is the outcome of the variable selection procedure? How do the betas look like and which information can you gain from that?

Response: We used min-max scaling. We added that to the manuscript, along with more information about the regression results as mentioned previously. The data dimensionality already appears in the manuscript in the experimental part. For the test cases considered, the model is not overly sensitive to hyperparameters – we added this to the manuscript. They were set by hand – the values used already appear in the manuscript. We added some interpretation of the results based on the estimated coefficients as mentioned above.

Comment: Table 1: The results are self-produced. How comparable are the results actually? What are the best results for the data set published in the literature so far?

Response: At the time the original manuscript was written, we were not aware of previous results for this particular version of the Cascaded Tanks dataset. There are a number of results for another version of the benchmark, which we did not use because of the lack of initial conditions and lack of information for the second state. However, this comment created the incentive to try the search again. We found a number that use the same version of the benchmark as we did, however comparisons are difficult because of the lack of consistent standards for validation in dynamic systems. We note that while all of these studies compared a new method against other methods as we did, none of them included results from other studies. Several do not make predictions against a validation set, reporting only results from training, which is not interesting. One of the studies that used a validation set performed very poorly. Of the two that remained, one (Zabiri et al., 2018, who implement a Support Vector Regression) used a means of validation that separated regions of extrapolation (test sets often overshoot the extremes in the state space encountered in training) from the rest of the test, while also including some training data in the test set. Since the extrapolation regions are where most of the error occurs, and including training data in the test set “pads the stats,” this makes quantitative comparison difficult.

The remaining paper (La Cava, et al., 2016) does not describe how the test set was chosen, but even if one assumes that there is a clear separation between test and training, comparisons are still difficult because of the different methods of testing. As described in the manuscript, we took the standard cross-validation approach more common in the machine learning community, which partitions the whole dataset (from a combination of two separate experiments provided by the benchmark) into five equal subsets, running five train-test studies using each of the partitions as test data in turn, then averaging the test statistics. We would be glad to include any study that used the same method in our tabulated results, however such a study seems not to exist. We did note in the text of the revision that we seem to easily outperform the new method reported in this paper -- Epigenetic Linear Genetic Programming (ELGP), in which an optimization procedure selects parametric terms to use in constructing a dynamic system – by significant margins: our MSE values are over an order of magnitude lower than ELGP, and approximately an order of magnitude lower than a very thinly parameterized recurrent neural network (NARX-NN), which outperformed ELGP.

Comment: Training and inference times: Tables summarizing and comparing the results would be nice.

Response: Though we report and compare training and inference times in the manuscript, simple numerical comparisons are again difficult because the methods we compare against are highly optimized, to include GPU acceleration. In our opinion, tabulated results are only appropriate for “apples-to-apples” comparisons, as many (likely most) readers will at least initially skim papers looking for tables and figures for quick comparisons. Any comparison with significant caveats is therefore best provided in the text of the manuscript. In this case, an at least equally compelling comparison regarding execution time is the number of parameters used in the model, which for us is two orders of magnitude lower than the RNNs, though higher than SINDy.

Comment: I think, a lot of research is going on around Gaussian processes at the moment. I was wondering why there is not so much literature cited (only 16 references). There are also recent papers concerning hyperparameter optimization for Gaussian processes (minimizing the negative log-likelihood).

Response: We have cited recent studies and included one new method (the OAK GP) in our comparisons. However as we note in the manuscript, most scalable GPs are better suited to static problems involving large datasets, rather than for dynamic systems. Since the original manuscript was written we have been made aware of a GP application to dynamic systems. This method uses applies a pair of Gaussian process kernels, one to link a high-dimensional observation space to a low-dimensional latent space, and another to describe dynamics on the latent space (Wang et al., 2005). While the dimensionality reduction is quite interesting it is not something we are addressing in this contribution, and the dynamic modeling seems to have more in common with the RNN methods than with our approach or SINDy. Nonetheless, since some may make the mistake of thinking we are using GPs in this sense, we included the reference.

Incidentally this paper, published at NeurIPS, cites 15 references.

Comment: I also want to draw your attention to publications by Potts and Schmischke, who make use of ANOVA-decompositions of functions in terms of Fourier partial sums (https://doi.org/10.1137/20M1354921 and follow-up studies) and using it within least-squares regression / Machine Learning applications.

Response: Thanks for the suggestion. This is an interesting application to which – like the dynamic GP – seems best suited to learning in situations where there i

---

## [Decision Letter · Decision Letter 1]

5 Jun 2024

PONE-D-23-14436R1Forward variable selection enables fast and accurate dynamic system identification with Karhunen-Loève decomposed Gaussian processesPLOS ONE

Dear Dr. Mebane,

Thank you for submitting your manuscript to PLOS ONE. After careful consideration, we feel that it has merit but does not fully meet PLOS ONE’s publication criteria as it currently stands. Therefore, we invite you to submit a revised version of the manuscript that addresses the points raised during the review process.

We look forward to receiving your revised manuscript.

Kind regards,

Yu Zhou

Academic Editor

PLOS ONE

Journal Requirements:

Reviewers' comments:

Reviewer's Responses to Questions

**Comments to the Author**

1. If the authors have adequately addressed your comments raised in a previous round of review and you feel that this manuscript is now acceptable for publication, you may indicate that here to bypass the “Comments to the Author” section, enter your conflict of interest statement in the “Confidential to Editor” section, and submit your "Accept" recommendation.

Reviewer #2: All comments have been addressed

Reviewer #3: All comments have been addressed

2. Is the manuscript technically sound, and do the data support the conclusions?

Reviewer #2: Yes

Reviewer #3: Yes

3. Has the statistical analysis been performed appropriately and rigorously? 

Reviewer #2: Yes

Reviewer #3: No

4. Have the authors made all data underlying the findings in their manuscript fully available?

Reviewer #2: Yes

Reviewer #3: Yes

5. Is the manuscript presented in an intelligible fashion and written in standard English?

Reviewer #2: Yes

Reviewer #3: Yes

6. Review Comments to the Author

Reviewer #2: I would like to thank the authors of the paper very much for their insightful responses to my numerous comments and the changes made.

Some of my questions were quite petty, for which I apologize. Despite my scientific proximity to the field, some of my questions arose from a certain lack of knowledge concerning some used basics details as well as pure interest. I hope that my comments have nevertheless been of help in revising the paper.

In my opinion, the authors have responded to all the comments made in the best possible way and have revised the paper accordingly. I recommend accepting the paper in this form.

Just a few small comments (that might be considered for the final revision for publication):

- in the abstract the complexity is stated, but it is not explained what N and P stand for (same page 3, line 18 and following. N=number of points, n=number of dimensions/features?)

- equation (4): one could write in the sum i=1 ... P, then it is immediately clear that P is the number of terms in the expansion

- the style of Figure 1 (size, visability, axis ticks) could be improved

Reviewer #3: The overall presentation of the work is good. Authors presented most of their funding in detail. However a small problem is on the definition of hyper parameters (HP), in some cases a form of optimization is applied to models and in some cases authors stated that “(Results were not particularly sensitive to the hyperparameter values”. It would be a good idea to present a detailed section or at least a table where the author provided full information on HP. HP is a crucial parameter that can significantly affect the overall performance of algorithms. Although authors mentioned that their models' HP are tuned, they did not provide a fully clear information on how they tuned it and whether the HP of counterpart methods are tuned optimally or not. Authors should add a section about this matter and novel approaches in this topic for the possible readers and clearly explain that if the hyper-parameters are not selected accurately or properly even the state-of-the-art methods would fail when compared with a tuned traditional model. 

7. PLOS authors have the option to publish the peer review history of their article (what does this mean?). If published, this will include your full peer review and any attached files.

Reviewer #2: No

Reviewer #3: No

---

## [Author Response · Author response to Decision Letter 1]

25 Jun 2024

See attached document, 'ResponseToReviewers2.docx'

---

## [Decision Letter · Decision Letter 2]

16 Aug 2024

Forward variable selection enables fast and accurate dynamic system identification with Karhunen-Loève decomposed Gaussian processes

PONE-D-23-14436R2

Dear Dr. Mebane,

We’re pleased to inform you that your manuscript has been judged scientifically suitable for publication and will be formally accepted for publication once it meets all outstanding technical requirements.

Kind regards,

Yu Zhou

Academic Editor

PLOS ONE

Additional Editor Comments (optional):

Reviewers' comments:

Reviewer's Responses to Questions

**Comments to the Author**

1. If the authors have adequately addressed your comments raised in a previous round of review and you feel that this manuscript is now acceptable for publication, you may indicate that here to bypass the “Comments to the Author” section, enter your conflict of interest statement in the “Confidential to Editor” section, and submit your "Accept" recommendation.

Reviewer #3: All comments have been addressed

2. Is the manuscript technically sound, and do the data support the conclusions?

Reviewer #3: Yes

3. Has the statistical analysis been performed appropriately and rigorously? 

Reviewer #3: Yes

4. Have the authors made all data underlying the findings in their manuscript fully available?

Reviewer #3: Yes

5. Is the manuscript presented in an intelligible fashion and written in standard English?

Reviewer #3: Yes

6. Review Comments to the Author

Reviewer #3: Authors had made an effort to improve the overall quality of the work. The reviewer has no further comments for this work. The current form of the work can be accepted for publication.

7. PLOS authors have the option to publish the peer review history of their article (what does this mean?). If published, this will include your full peer review and any attached files.

Reviewer #3: No

---

## [Editor Report · Acceptance letter]

11 Sep 2024

PONE-D-23-14436R2 

PLOS ONE

Dear Dr. Mebane, 

I'm pleased to inform you that your manuscript has been deemed suitable for publication in PLOS ONE. Congratulations! Your manuscript is now being handed over to our production team.

Kind regards, 

on behalf of

Dr. Yu Zhou 

Academic Editor

PLOS ONE